# New Insights on Liver-Directed Therapies in Hepatocellular Carcinoma

**DOI:** 10.3390/cancers15245749

**Published:** 2023-12-08

**Authors:** Christina G. Dalzell, Amy C. Taylor, Sarah B. White

**Affiliations:** 1Department of Radiology and Medical Imaging, Division of Vascular and Interventional Radiology, University of Virginia Health System, Charlottesville, VA 22903, USA; 2Department of Radiology, Division of Vascular and Interventional Radiology, Medical College of Wisconsin, Milwaukee, WI 53226, USA

**Keywords:** hepatocellular carcinoma, chemoembolization, radioembolization, ablation, interventional radiology, interventional oncology, immunotherapy

## Abstract

**Simple Summary:**

Hepatocellular carcinoma is the most common type of liver cancer and affects a significant number of people worldwide. There are multiple treatment options available to patients depending on their functional status and characteristics of their tumor. Interventional radiologists perform many of the liver directed therapies, including ablation and intra-arterial catheter-based therapies such as transarterial chemoembolization and transarterial radioembolization. In this article, we describe these liver-directed therapies in the context of current treatment recommendations as well as ongoing research into new therapies and the combination of therapies.

**Abstract:**

The incidence of hepatocellular carcinoma (HCC) has been increasing over the past decades, but improvements in systemic and locoregional therapies is increasing survival. Current locoregional treatment options include ablation, transarterial chemoembolization (TACE), transarterial radioembolization (TARE), and stereotactic body radiotherapy (SBRT). There is ongoing research regarding the combination of systemic and local therapies to maximize treatment effect as well as in new non-invasive, image-guided techniques such as histotripsy. There is also active research in optimizing the delivery of therapy to tumors via nanostructures and viral-vector-mediated gene therapies. In many cases, patients require a combination of therapies to achieve tumor control and prolong survival. This article provides an overview of the most common liver-directed therapies for HCC as well as insight into more recent advances in personalized medicine and emerging techniques.

## 1. Introduction

Hepatocellular carcinoma (HCC) is the sixth most common type of cancer, the most common primary liver cancer, and the third leading cause of cancer death [1]. The development of HCC is related to cirrhosis in over 90% of cases, though the etiology of underlying liver disease varies globally. In Western countries, the most common causes are alcohol-related liver disease, metabolic dysfunction-associated steatotic liver disease (MASLD), and viral hepatitis including hepatitis B (HBV) and C (HCV) viruses. Viral hepatitis infections and aflatoxin exposure are the most common causes worldwide [2].

Routine screening for HCC with ultrasound (US) and alpha fetoprotein (AFP) serum levels are recommended in patients with cirrhosis and high-risk patients without cirrhosis but with chronic HBV [2]. Studies have shown that only 40% of patients are diagnosed with early HCC, which has important implications for prognosis and treatment options [3]. If the patient is a candidate for surgical resection or transplantation, the 5-year survival is over 70% [4]. However, because the majority of patients are diagnosed at late stages, significant developments in additional treatment options for HCC have been developed. The Barcelona clinic liver cancer (BCLC) staging system is the most widely accepted staging system and includes a patient’s performance status, degree of liver dysfunction, and tumor status [2,5]. The Eastern Cooperative Oncology Group (ECOG) performance status scale is a five-point scale used to evaluate a patient’s functional status [6]. Liver dysfunction was previously classified by the Child–Pugh score, but the BCLC guidelines were updated in 2022 to reflect a binary classification of liver dysfunction into decompensated versus compensated liver disease [7]. Decompensated liver disease accounts for the presence of jaundice, ascites, or hepatic encephalopathy (HE) [7]. The BCLC staging system also includes information on tumor status, which includes the tumor size, number of tumors, extent of vascular and/or lymphatic invasion, and the presence of metastases [2]. With this information, patients can be classified into very early (0), early (A), intermediate (B), advanced (C), and terminal (D) stages with corresponding estimated prognoses and treatment pathways [7].

Treatment options for HCC can be broadly characterized into medical, surgical, or local therapies with either curative or non-curative intent. Liver transplantation, tumor resection, and ablation are considered curative in patients with HCC [7]. For patients who are not candidates for curative therapy, other liver-directed therapies have been developed to prolong survival or downstage patients to surgical resection or transplantation, including percutaneous ablation, stereotactic body radiotherapy (SBRT) and transarterial therapies.

Various ablative techniques including microwave (MWA), radiofrequency (RFA), and cryoablation were first developed in the late 20th century, based on the principle of applying heating or cooling directly to the tumor to incite cellular necrosis [8]. Image-guided percutaneous tumor ablation developed as an alternative to surgical ablation in poor surgical candidates, especially as cross-sectional imaging improved, though large tumor size or proximity to critical structures can limit the application of these modalities [8].

The success of early catheter directed intra-arterial therapies for HCC is based on the principle of the liver’s dual blood supply. The portal vein accounts for approximately 75% of the liver’s blood supply while the hepatic artery accounts for 25% [9,10]. However, HCC obtains up to 90% of its blood from the hepatic artery which allows for intra-arterial administration of therapy that occludes the tumor supplying arteries, resulting in tumor necrosis while sparing normal liver parenchyma, which preferentially derives blood supply from the portal vein [9]. This forms the basis for transarterial chemoembolization (TACE). Transarterial radioembolization (TARE) is another intra-arterial therapy; however, with TARE, radiation-carrying micron-sized beads are delivered to the tumor and cause cell death by radiation damage. Both TACE and TARE are important pillars of intermediate stage HCC therapy.

The first transarterial therapies for HCC were performed in Japan starting in the late 1970s, with early studies showing improved survival when compared to traditional systemic chemotherapy regimens [11,12,13]. In 2002, a randomized controlled trial by Llovet et al. demonstrated improved survival in intermediate stage patients who underwent chemoembolization when compared to those receiving conservative treatment alone [14].

Researchers first began experimenting with the intra-arterial delivery of the radioisotope Yttrium-90 (Y90) as tumoricidal therapy in the 1960s, first in a canine model and then in humans [9,10,15,16]. The technique was developed further, and the development of glass- and resin-coated microspheres in the late 1980s-early 1990s led to the first phase I clinical trials of transarterial radioembolization (TARE), showing this to be a safe and effective treatment for HCC [16,17].

The role of interventional radiology in the management of HCC has quickly evolved over the past few decades. Interventional radiologists remain positioned to offer a variety of image-guided techniques that improve the clinical outcome of early- to late-stage HCC. Interdisciplinary tumor boards with radiologists, interventional radiologists, transplant and surgical oncologists, hepatologists, radiation oncologists, and medical oncologists remain necessary to ensure that each treatment pattern is personalized to the patient and tumor characteristics. This article will provide an overview of the most common current liver directed therapies in HCC, including percutaneous ablation, TACE, and TARE, as well as insight into more recent advances in personalized medicine and emerging techniques.

## 2. Overview of Current Liver Directed Therapies for HCC

### 2.1. Percutaneous Ablation

Percutaneous ablation is a curative option for patients with very early stage (BCLC 0) or early stage (BCLC A) HCC tumors < 3 cm who are not otherwise candidates for liver transplantation or resection [7]. Various techniques have been developed including RFA, MWA (Figure 1), cryoablation, and irreversible electroporation (IRE). These techniques require the tumor to be in a favorable location with a margin of normal liver tissue between the tumor and a critical structure. RFA is a heat-based technique that was developed alongside the electrocautery knife used by surgeons [8]. RFA involves creating a closed-loop-circuit with an RF electrode, generator, and grounding pads to produce an alternating current that agitates ions in the tissue leading to heat generation [8]. RFA is susceptible to the heat sink effect whereby adjacent vessels can dissipate heat and thus affect the ability to generate the necessary temperature for tumor necrosis, which is one of the pitfalls of this technique [8]. When compared with surgical resection, RFA has demonstrated similar local control and long-term survival with a significantly lower rate of complications as well as a shorter length of hospital stay [18,19,20].

MWA involves producing high frequency oscillating electromagnetic fields which lead to the rapid oscillation of water molecules and the generation of heat. This technique generates heat with a wider zone of the primary heating and is less susceptible to the heat sink effect than RFA. Therefore, higher temperatures can be achieved with a shorter treatment duration, in some cases providing benefit over RFA [8]. MWA has been shown to result in better 5-year survival when compared with RFA for tumors larger than 3.5 cm [21].

Cryoablation is based on the principle that cold temperatures applied to tissue causes intra- and extra-cellular ice crystal formation which leads to cell death. This technique was originally studied and deployed using liquid nitrogen, but now utilizes argon-based cooling [8]. The benefits of cryoablation over heat-based ablation is the ability to monitor the ablation zone in real time via “ice ball” formation on cross-sectional imaging and elicits far less pain due to the anesthetic effect of cooling on nerves [22]. When compared with RFA, cryoablation has shown similar cancer-specific survival and overall survival rates in a large population-based study using the surveillance, epidemiology, and end results (SEER) HCC database and similarly, when patients were paired in a propensity-matched analysis [23,24]. However, cryoablation has been shown to be associated with a higher rate of serious complications including “cryoshock”, a cytokine mediated systemic inflammatory process, and disseminated intravascular coagulation (DIC), which limits its application in certain patients [25].

Irreversible electroporation (IRE) is a non-thermal technique which involves high intensity electrical pulses that damage cell membranes while preserving the extracellular matrix, allowing for more rapid tissue healing [26]. This technique allows for precise control of the location of damage. Hence, it can be used in tumors with close proximity to blood vessels or bile ducts [26]. This technique involves applying more applicators and precise alignment to deliver the desired effect to the tumor [25]. Compared to RFA, IRE results in a smaller axial diameter and area of ablation at 3 months and 1-year post-procedure with no difference in local tumor progression [26]. Due to the risk of alterations in general ion transport in the body with this technique, general anesthesia may be required [25].

Overall, ablation remains an important locoregional therapy for patients with early stage HCC who would not otherwise be surgical candidates. The adverse events possible with ablation include bleeding, tumor seeding, abscess formation, and biliary complications [25]. However, the benefits and drawbacks of each type of ablation can be weighed by the interventional radiologist to guide the treatment of tumors while minimizing damage to normal parenchyma.

### 2.2. Transarterial Chemoembolization (TACE)

TACE is the mainstay of treatment in intermediate stage (BCLC-B) HCC. Conventional TACE (cTACE) is performed with a combination of ethiodized oil and chemotherapy, in the US most commonly doxorubicin, either alone or in combination with mitomycin and cisplatin, followed by an embolic agent such as polyvinyl alcohol (PVA) or a gelatin sponge (Figure 2). TACE with drug-eluting beads (DEB-TACE) uses doxorubicin-loaded microspheres as a simultaneous drug-delivery and embolic agent. In the PRECISION V trial, the two techniques were compared, and results demonstrated a decrease in systemic side effects with DEB-TACE with similar survival rates [27]. Another study by Li et al. similarly showed a decrease in complications and prolonged interval between treatments with DEB-TACE [28]. In a more recent study, the PRESIDENT trial, cTACE was again compared to the more modern techniques of DEB-TACE [29]. The results demonstrated that the cTACE cohort had higher rates of complete response at 1- and 3-month time points (84.2%, 75.2%) versus the DEB-TACE group (35.7%, 27.6%). However, the cTACE group had significantly higher rates of adverse events attributed to post embolization syndrome including fever, fatigue, malaise, abdominal pain, anorexia, and lab abnormalities including elevated bilirubin, AST, ALT, and hypoalbuminemia [29].

TACE has demonstrated significant survival benefit for patients with HCC over supportive care alone [14,30]. However, there are a subset of patients who do not tolerate the side effects or whose tumors do not show a significant response. There has been research interest in identifying the patient and tumor characteristics that make these tumors “unsuitable” or refractory to TACE. Historically, a TACE failure was defined as stage migration from BCLC B to C. However, now the definition has been expanded to include patients who do not show objective response after two TACE sessions with the goal of initiating systemic therapy earlier and preserving liver function [3,31].

### 2.3. Transarterial Radioembolization (TARE)

TARE was originally offered as an alternative to TACE in patients with portal vein thrombosis (PVT), a common complication of HCC [10,32]. This is based on the principle that TARE is non-embolic and thus can be used safely in cases of PVT and does not preclude future treatment with TACE [10]. TARE involves the radioisotope Yttrium-90 (^90^Y) which undergoes beta decay and has a tissue penetration of 2.5 to 11 mm [9]. In the LEGACY trial, patients with unresectable HCC who underwent TARE demonstrated an objective response rate of 88.3% with a 3-year overall survival of 86.6% [33]. Other studies have focused on comparing the efficacy and outcomes of TACE versus TARE in early and intermediate stage HCC [34]. In the PREMIER trial, patients who underwent TARE had significantly longer time to progression (TTP) (>26 months) versus patients who underwent TACE (8.2 months; *p* = 0.0012) with similar median survival time [34]. In the TRACE II trial, patients with intermediate stage HCC were randomized to either TARE or DEB-TACE [35]. The primary endpoint of time to tumor progression was significantly longer in the TARE versus DEB-TACE group (17.1 versus 9.5 months; *p* = 0.002). Median overall survival was also significantly longer in the TARE group (30.2 versus 15.6 months, *p* = 0.006). Both groups had similar rates of adverse events and 30-day mortality [35]. Other studies have shown better tolerated side effects, decreased length of hospital stays, and fewer treatment sessions required with TARE [36,37].

Variations in TARE include radiation segmentectomy and lobectomy [10]. In patients in which resection or ablation is not feasible, radiation segmentectomy can offer an ablative level treatment dose to lesions confined to less than two segments while sparing the remaining normal liver parenchyma [10] (Figure 3). The RASER trial demonstrated that radiation segmentectomy in patients with a solitary HCC tumor in an unfavorable location for ablation resulted in high rates of complete response (24/29 patients, 83%) with low rates of adverse events [38]. In the DOSISPHERE-01 study, patients with unresectable HCC were randomized to either standard dosimetry (120 ± 20 Gy) to the diseased lobe or personalized dosimetry (>205 Gy) to the index lesion. Patients in the personalized dosimetry group had significantly higher rates of objective response to therapy with similar rates of adverse effects [39]. Radiation lobectomy is yet another application of ^90^Y that involves administering a higher dose to the diseased lobe of the liver, with the goal of inducing tumor death and atrophy of the diseased lobe with resultant compensatory hypertrophy of the normal lobe. This concept can allow patients who would not otherwise be candidates for surgical resection due to a small future liver remnant (FLR) to be able to undergo resection [10].

### 2.4. Other Locoregional Therapies

Hepatic artery infusion chemotherapy (HAIC) and stereotactic body radiotherapy (SBRT) are locoregional techniques that have also been described in unresectable HCC [40]. HAIC has been historically used in patients with unresectable HCC who have progressed after TACE or patients with portal vein thrombosis or large infiltrative tumors [40]. Historically, the combination of 5-FU and cisplatin are drugs utilized [40]. In a recent FOHAIC-1 trial, the combination of oxaliplatin and 5-FU was compared to the systemic agent sorafenib [41]. Patients who received the combination therapy demonstrated significantly longer survival and were more likely to have their tumors downgraded. Hence, there is growing interest in using HAIC as an alternative for patients with advanced stage HCC who are not candidates for other locoregional therapies or who have not responded to systemic therapy [40].

External beam radiotherapy originally had a limited role in the treatment of HCC due to the risk of radiation-induced liver injury [42]. However, technological advancements have enabled higher doses to be delivered more precisely in SBRT. This form of radiation therapy has been shown to induce immunogenic cell death, which can have synergistic effects when combined with other therapies [42]. It is generally used in early stage HCC in patients who are not candidates for resection or ablation or as a palliative therapy in locally advanced or metastatic HCC [42]. A study by Sapisochin et al. demonstrated similar survival and postoperative complication rates in patients with HCC who were bridged to transplant with either RFA, TACE, or SBRT [43]. Further research is needed on the application of this therapy versus other locoregional therapies and its role in the HCC treatment paradigm.

## 3. Recent Advances and Innovations in Liver Directed Therapies for HCC

### 3.1. Precision Medicine and Personalized Approaches

HCC arises from a background of liver cirrhosis in the majority of cases. As there are vastly different etiologies for cirrhosis, there is interest in better understanding the differences in tumors on the molecular level that can be used to maximize treatment effect. The application of genome sequencing in HCC has been limited by the fact that many patients do not undergo biopsy prior to treatment as the combination of certain imaging findings with elevated AFP is highly sensitive and specific for HCC, obviating the need for tissue diagnosis [44]. Circulating tumor DNA (ctDNA), circulating tumor cells (CTC), and circulating cell-free microRNA (miRNA) are all serological markers of HCC that have been studied as alternatives to traditional tissue biopsy [44]. These “liquid biopsy” techniques have been shown to have a high sensitivity and specificity for HCC. In China, there is a detection kit using circulating miRNA that is used to diagnose HCC [45].

Genetic sequencing is also being used to predict tumor recurrence, metastasis, and response to locoregional therapies [41]. The ALB1 mutation has been associated with tumor recurrence and has been found in ctDNA at the time of relapse [46]. In terms of response to TACE, HCC tumors with a mutated NRF2 pathway have been identified as particularly resistant to hypoxia and thus show rapid progression despite treatment [47]. This mutation is found in up to 14% of patients, which highlights the importance of creating drugs that target this pathway or finding alternative treatment options for patients with this particular tumor mutation [47].

Molecular profiling remains in the pre-clinical phase in the United States. However, there is great potential in utilizing molecular tumor information to personalize treatment options for patients beyond the BCLC guidelines.

### 3.2. Combination Therapies

Systemically delivered targeted therapies have been approved for the treatment of advanced-stage HCC. Sorafenib and Lenvatinib are tyrosine kinase inhibitors (TKIs) that are involved in tumor angiogenesis and were the first FDA-approved systemic targeted therapies [48]. More recently, the combination of the monoclonal antibodies Atezolizumab and Bevacizumab (Atezo-Bev) has become the first line agent in unresectable, advanced stage HCC and has provided survival benefit when compared to sorafenib [7,49]. The IMBRAVE 050 is a phase III trial comparing adjuvant Atezo-Bev in patients at high risk of HCC recurrence following ablation or resection versus active surveillance [50]. Results demonstrated statistically significant improvement in the primary endpoint of recurrence free survival in the group treated with adjuvant Atezo-Bev [51]. Atezolizumab is an immune checkpoint inhibitor, while Bevacizumab is a VEGF inhibitor that targets tumor angiogenesis [49]. Studies have shown that the hypoxic environment that is created post-arterial embolization can stimulate VEGF expression and angiogenesis [48]. Hence, there is interest in combining these systemic therapies with intra-arterial embolization to enhance the tumoricidal effect [48]. The HIMALAYA trial used a combination of the checkpoint inhibitors tremelimumab (CTLA-4 inhibitor) and durvalumab (anti-PDL-1) in an infusion regimen termed STRIDE (single tremelimumab, regular interval durvalumab). The STRIDE regimen demonstrated significantly improved overall survival when compared with Sorafenib [52]. Similarly, the combination of the immune checkpoint inhibitor camrelizumab with the TKI rivoceranib in the CARES-310 phase III trial showed significant increase in PFS and OS when compared to Sorafenib [53]. These combinations have become other first line systemic therapies in advanced HCC.

The combination of systemic therapy with TACE has been an active area of research (Appendix A). There have been multiple randomized trials which compared sorafenib in combination with cTACE or DEB-TACE to TACE alone [51,52,53,54]. The SPACE and TACE 2 trials failed to show improvement in the primary endpoint of TTP or progression free survival (PFS) with the combination [54,55]. However, the TACTICS trial utilized the secondary endpoint of time to “un-TACEable” progression to maximize the duration of sorafenib treatment and more closely model what actually happens in clinical practice [56,57]. This study demonstrated significant improvement in PFS in patients treated with the combination of sorafenib and TACE versus TACE alone. The overall survival was higher in patients who received the combination, but this difference was not statistically significant [57]. Another study by Xia et al. and the phase III trial LAUNCH compared lenvatinib with DEB-TACE versus lenvatinib alone, which showed improved overall survival (OS) and PFS in the patients who received combination therapy [58,59].

Checkpoint inhibitors have also been studied in combination with locoregional therapies. Nivolumab and pembrolizumab are both PD-1 inhibitors involved in T-cell suppression that have recently been approved for the treatment of HCC. Studies have begun to combine this with a tyrosine kinase inhibitor and TACE with promising results [60,61,62]. A current phase II clinical trial is evaluating the combination of Atezo-Bev and TACE or TARE with encouraging objective tumor response rates of 61.9% [63,64]. A phase III trial EMERALD-1 is currently ongoing which is focused on comparing durvalumab monotherapy with either DEB-TACE or cTACE followed by Durvalumab or a combination of Durvalumab and Bevacizumab versus TACE alone (NCT03778957). The EMERALD-3 study is another ongoing phase III trial assessing the efficacy and safety of durvalumab + tremelimumab + TACE with or without Lenvatinib compared with TACE alone (NCT05301842). Thus, the combination of direct macroscopic blockage of tumorigenesis with TACE and cellular-level inhibitors of angiogenesis and immune modulators offer promising synergistic effects that will continue to be studied.

The combination of locoregional therapies is also an area of interest. In a meta-analysis, the combination of TACE and RFA compared to surgical resection showed no difference in overall survival but reduced complications in the combination therapy group [65]. The ethiodized oil retention within HCC following cTACE can also be used as a marker to guide percutaneous ablation (Figure 4). Stereotactic body radiotherapy (SBRT) is currently being studied in combination with TACE with promising results. In a propensity-scored matched analysis by Wong et al., patients who received the combination of SBRT + TACE had significantly higher one- and three-year overall survival and improved radiological disease control [66]. There is also interest in using SBRT in cases of local relapse following TACE. A phase III trial was closed early due to slow accrual, but results showed superior local control of SBRT versus repeat TAE/TACE [67]. SBRT remains a promising non-invasive treatment for unresectable HCC. Further research is needed to define its role within the HCC treatment paradigm.

### 3.3. Targeted Drug Delivery

The application of nanotechnology in drug delivery is an exciting frontier for interventional radiologists. The tumor microenvironment of HCC has been well studied and multiple clinical trials have focused on various nanostructures and their application in HCC [68]. Nanostructures are <100 nm and can be developed to control delivery of drugs or other therapies. One of the limitations of cTACE is that the mixture of ethiodized oil, a hydrophobic agent, and doxorubicin, a hydrophilic agent, is unstable and can lead to a rapid release of the drug which has less tumoricidal effect [69]. In comparison, nanostructures can be deployed as controlled delivery systems to directly deliver the therapeutic agent to the tumor microenvironment; for example, in the setting of hypoxia or acidosis in which systemic chemotherapy can be ineffective [69]. Multiple preclinical trials in animal and cell models are focused on encapsulating chemotherapeutic agents into nanoparticles, with promising results in inhibiting tumor growth [70]. 

In the ThermoDox study, lyso-thermosensitive liposomal doxorubicin (LTLD, ThermoDox) was doxorubicin contained within a heat-sensitive liposome that was administered intravenously during percutaneous RFA [71]. In the phase III HEAT trial, the combination failed to reach the primary end point of PFS [71]. The more recent OPTIMA trial was also stopped early due to futility [72]. Further research has focused on viral-vector mediated gene delivery, some of which can be administered intra-arterially [73]. Adenovirus is the most commonly used viral vector. Multiple early phase clinical trials have used this vector with p53, a tumor suppressor gene that is mutated in up to 40–50% cases of HCC [73]. Multiple clinical trials remain in the recruiting phase with the goal of delivering functional copies of this gene to induce apoptosis in cancer cells [73].

### 3.4. Other Emerging Techniques

Histotripsy is a noninvasive, non-thermal image-guided ablation technique that has been studied in animal models of HCC with promising early results in human clinical trials. It uses short bursts of ultrasound waves with higher peak amplitudes to generate acoustic cavitation using endogenous gas in tissues [74]. This movement of microbubbles results in mechanical destruction of cells. This technique creates well-defined borders, which provides benefits over the heat-based ablation techniques described previously [74]. A multicenter phase I trial (THERESA study) was completed in 2018–2019 with technical success defined as visualization of an ablation zone on post-procedure MRI [75]. A phase I/II trial (#HOPEFORLIVER) using the HistoSonics system is ongoing [76]. Histotripsy is a promising technique in the realm of locoregional therapies.

## 4. Patient Selection

With all the data and new advances in HCC therapies, treatment selection can be challenging. Therefore, it is crucial to have a multidisciplinary discussion with a patient -centered approach to determine treatment algorithms for each patient. The treatment algorithms also can vary by geographic location, due to availably of devices and drugs for treatment. Notwithstanding, the approach needs to consider tumor location, which is often not addressed in current guideline recommendations, but can be critical in safety and effectiveness of many locoregional therapies. In very early and early stage HCC (BCLC 0 and A), the goal is curative therapy [7], including surgical resection or liver transplantation for surgical candidates and ablation for non-surgical candidates. However, if the tumor is located centrally and adjacent to a critical structure that would preclude ablation, TACE, TARE, or SBRT should be strongly considered to mitigate the risk of complications. In the intermediate stage (BCLC B), multinodular HCC with preserved liver function either transarterial therapies (TACE/TARE) or SBRT can be employed. The choice of therapy depends on several factors, including the goal of therapy. Therapies can be performed with the goal of downstage or bridge patients to liver transplant, for potential curative intent in the case of TARE, or for palliation [7]. The other factors to consider are number and distribution of lesions, presence of vascular invasion, presence of arterioportal or hepatopulmonary shunting, hepatic reserve, prior therapies, and overall performance status. For patients with diffuse infiltrative bilobar HCC who are not candidates for locoregional therapy and for advanced and terminal stage (BCLC C and D) patients with portal invasion and/or extrahepatic spread, systemic therapy is the mainstay of treatment [7]; however, combination therapy may be beneficial, with forthcoming data. 

## 5. Conclusions

Image-guided locoregional therapies performed by interventional radiologists remain important in the treatment paradigm for patients with HCC. As the incidence of HCC continues to rise, there is significant interest in the evolution of current techniques and the development of new ones. In 2014, the Hong Kong liver cancer (HKLC) staging system was developed to better stratify patients in Asia with HCC, the majority of which was related to HBV with preserved liver function. These guidelines, similar to the BCLC ones, offer prognostic and therapeutic recommendations and are more aggressive with regard to surgical resection [77,78]. Both the BCLC and HKLC guidelines demonstrate significant improvement in survival for patients with HCC with a median survival of over 5 years in the early stages (0 and A), 2.5 years in intermediate stage (B), and 2 years in advanced stage HCC [7]. In comparison, for untreated HCC, median survival was previously 25 months in early stage (A), 10 months in intermediate stage (B), and 7 months in advanced [79]. Hence, there have been significant improvements in therapy over the past decades for patients with unresectable HCC. The guidelines also outline the possible need for multiple types of therapy, including systemic and locoregional, to achieve tumor control and prolong survival. With all of these options, ongoing research will help delineate optimal timing and which locoregional and systemic therapies should be offered to patients with HCC.

## Figures and Tables

**Figure 1 cancers-15-05749-f001:**
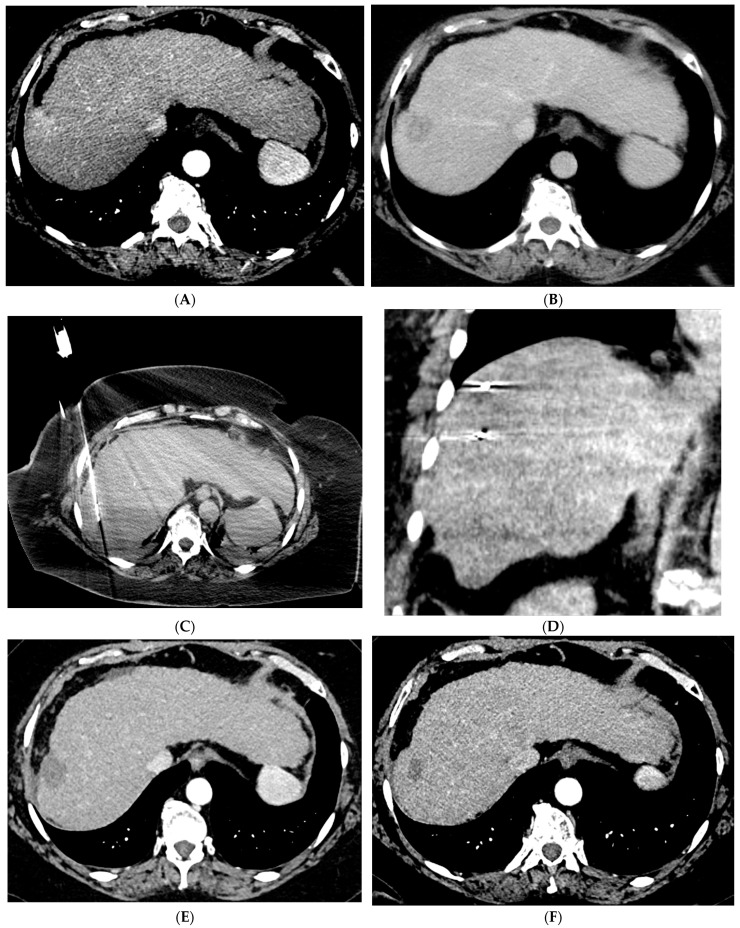
66-year-old woman with cirrhosis secondary to autoimmune hepatitis and HCC, treated with microwave ablation. (**A**) Arterial phase MRI showing a 3.5 cm arterially enhancing mass in the periphery of segment 7. (**B**) Delayed phase MRI demonstrating washout of the mass. Intraprocedural axial (**C**) and coronal (**D**) CT images demonstrating probe placement bracketing the mass. (**E**) Arterial phase MRI one month post ablation showing no residual viable tumor. (**F**) Arterial phase MRI one year post ablation demonstrating no residual or recurrent viable tumor and involution of the ablation cavity.

**Figure 2 cancers-15-05749-f002:**
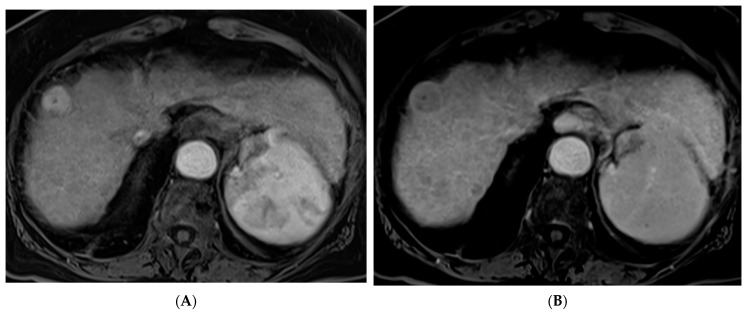
88-year-old woman with HCC on a background of cirrhosis secondary to autoimmune hepatitis treated with conventional TACE. (**A**) Arterial phase MRI showing a 2.2 cm arterially enhancing lesion in hepatic segment 8. (**B**) Delayed phase MRI showing washout of the lesion. (**C**) Intraprocedural CTA demonstrating supply to the hypervascular lesion from the segment 8 arterial branch. (**D**) Post procedure non-contrast CT demonstrating the ethiodized oil deposition within the tumor and surrounding segment 8 parenchyma. (**E**) Follow up arterial phase MRI one month after treatment demonstrating no arterial enhancement within the treated lesion consistent with complete response.

**Figure 3 cancers-15-05749-f003:**
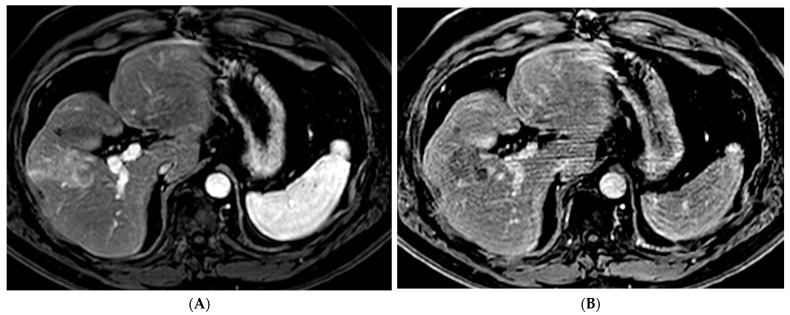
73-year-old man with non-cirrhotic HCC treated with ^90^Y radiation segmentectomy. (**A**) Arterial phase MRI showing a 4 cm arterially enhancing lesion in hepatic segment 5. (**B**) Delayed phase MRI showing washout of the lesion. (**C**) Intraprocedural CTA demonstrating supply to the posterior aspect of the mass from one of the segment 5 branch arteries. (**D**) Intraprocedural CTA demonstrating supply to the anterior aspect of the mass from a separate segment 5 branch artery. The Y^90^ dose was delivered as a split dose between these two arteries. (**E**) Follow up arterial phase MRI 3 months after treatment demonstrates wedge-shaped post-treatment changes in hepatic segment 5 with expected parenchymal enhancement and capsular retraction with no residual enhancement of the targeted tumor. (**F**) Follow up delayed phase MRI demonstrates no wash-out of the enhancing parenchyma to suggest residual viable tumor, consistent with complete response.

**Figure 4 cancers-15-05749-f004:**
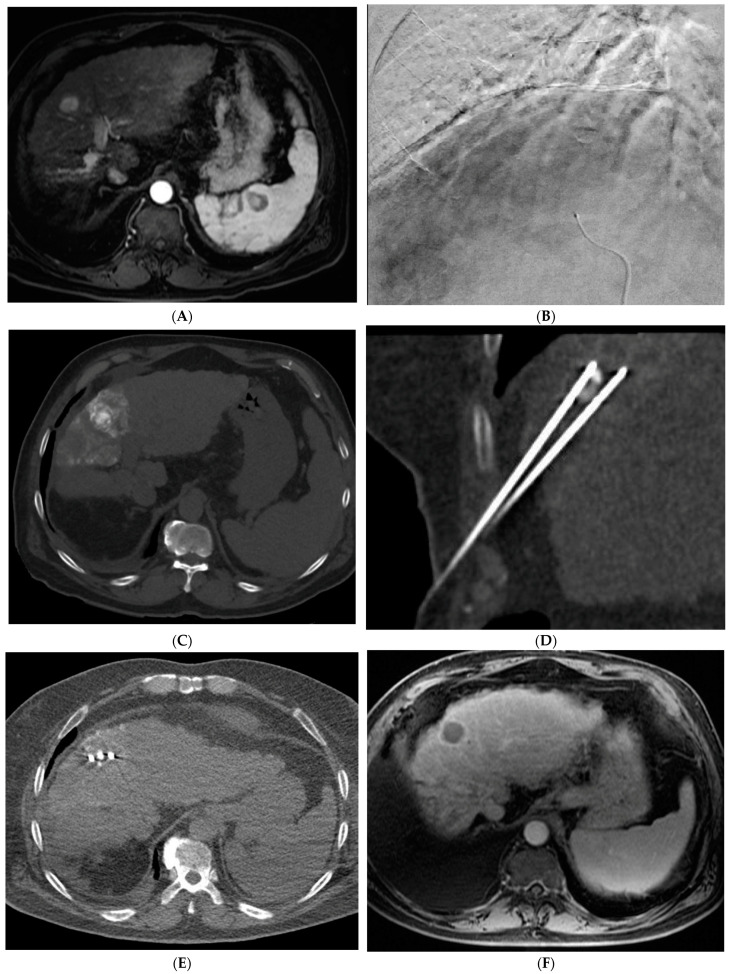
72-year-old man with HCV cirrhosis and HCC treated with combined cTACE and cryoablation as a bridge to transplant. (**A**) Arterial phase MRI demonstrates a 2.2 cm segment 4a hypervascular lesion. (**B**) Digital subtraction angiography images during cTACE procedure demonstrate that the tumor was supplied by both the segment 4a branch vessel (image shown) and segment 2 artery (not shown). cTACE was performed using doxorubicin 50 mg, cisplatin 100 mg, and mitomycin 10 mg followed by PVA 150–250 micron particles. (**C**) A non-contrast CT scan performed on post-operative day 1 demonstrates a heterogeneous uptake of ethiodized oil within the tumor. Cryoablation was then performed using two probes. Coronal and axial CT images from the procedure demonstrate the probes adjacent to the ethiodized oil staining (**D**) and the ice ball (**E**). (**F**) A follow up contrast enhanced MRI in the arterial phase demonstrates no residual viable tumor.

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
