# Peer review of "New Insights on Liver-Directed Therapies in Hepatocellular Carcinoma"

_cancers, 2023, doi:10.3390/cancers15245749_

Round 1

Reviewer 1 Report

Comments and Suggestions for Authors

This manuscript explores the landscape of liver-directed therapies for hepatocellular carcinoma (HCC), emphasizing current approaches within three primary modalities: percutaneous ablation, transarterial chemoembolization (TACE), and transarterial radioembolization (TARE), recent advances, including precision medicine and personalized approaches, combination therapies, targeted drug delivery, and emerging techniques like histotripsy. In summary, the paper provides an insightful review of current HCC therapies, recent advancements, and the promise of emerging approaches. However, there are some aspects that could be explored in more detail:

1 The authors could provide more insights into the specific patient characteristics that align with particular therapies. This would encompass the factors that guide treatment decisions, especially in light of recent progress in precision medicine and personalized approaches.

2 Considering that the HKLC staging system has been developed to refine patient stratification in Asia, it would be valuable to explore how these staging systems are tailored to different regions. Investigating the challenges and disparities in HCC management and treatment across diverse geographical regions would shed light on these critical aspects.

Author Response

Response to Editors and Reviewers Comments:

We would like to thank the editors and reviewers for their constructive assessment of our work and helpful recommendations. Please find our responses to your comments below. All comments are addressed in an individual manner.

Reviewer 1:

This manuscript explores the landscape of liver-directed therapies for hepatocellular carcinoma (HCC), emphasizing current approaches within three primary modalities: percutaneous ablation, transarterial chemoembolization (TACE), and transarterial radioembolization (TARE), recent advances, including precision medicine and personalized approaches, combination therapies, targeted drug delivery, and emerging techniques like histotripsy. In summary, the paper provides an insightful review of current HCC therapies, recent advancements, and the promise of emerging approaches. However, there are some aspects that could be explored in more detail:

  1. The authors could provide more insights into the specific patient characteristics that align with particular therapies. This would encompass the factors that guide treatment decisions, especially in light of recent progress in precision medicine and personalized approaches.

We thank the reviewers for their comments, and have updated the text to include this paragraph, just prior to the conclusion.

“With all the data and new advanced in HCC therapies, treatment selection can be challenging. Therefore, it is crucial to have a multidisciplinary discussion with a patient -centered approach to determine treatment algorithms for each patient. The treatment algorithms also can vary by geographic location, due to availably of devices and drugs for treatment. Notwithstanding, the approach needs to consider tumor location, which is often not addressed in current guideline recommendations, but can be critical in safety and effectiveness of many locoregional therapies. In very early and early-stage HCC (BCLC 0 and A), the goal is curative therapy (7), including surgical resection or liver transplantation for surgical candidates and ablation for non-surgical candidates. However, if the tumor is located centrally and adjacent to a critical structure that would preclude ablation, TACE, TARE or SBRT should be strongly considered to mitigate the risk of complications. In intermediate stage (BCLC B) multinodular HCC with preserved liver function either transarterial therapies (TACE/TARE) or SBRT can be employed. The choice of therapy depends on several factors, including the goal of therapy. Therapies can be performed with the goal to downstage or bridge patients to liver transplant, for potential curative intent in the case of TARE, or for palliation (7). The other factors to consider are number and distribution of lesions, presence of vascular invasion, presence of arterioportal or hepatopulmonary shunting, hepatic reserve, prior therapies, and overall performance status. For patients with diffuse infiltrative, bilobar HCC who are not candidates for locoregional therapy and for advanced and terminal stage (BCLC C and D) patients with portal invasion and/or extrahepatic spread, systemic therapy is the mainstay of treatment (7); however, combination therapy may be beneficial, with data forthcoming.”

  1. Considering that the HKLC staging system has been developed to refine patient stratification in Asia, it would be valuable to explore how these staging systems are tailored to different regions. Investigating the challenges and disparities in HCC management and treatment across diverse geographical regions would shed light on these critical aspects.

We thank the reviewers for this comment; however, we believe that comparing the staging systems across the globe is beyond the scope of this review article.  We have however added the following statement to address the issue of non-standardized approaches globally. “The treatment algorithms also can vary by geographic location, due to availably of devices and drugs for treatment.”

Reviewer 2 Report

Comments and Suggestions for Authors

Dear Authors,

We have carefully completed the preview of this manuscript. This is a relatively complete review on the targeted therapy of HCC. Here are a few recommendations for this manuscript. At the part of Overview of Current Liver Directed Therapies for HCC. Firstly, because of the significant achievements in HAIC therapy in intermediate-stage HCC in recent years, I would suggest to add relevant content. (Please refer to: doi:10.1200/JCO.21.01963). Secondly, please consider to discuss the combination therapy of TACE and ablation. Thirdly, SBRT remains an important treatment option for HCC, please consider to add relevant content. At the part of Recent Advances and Innovations in Liver Directed Therapies for HCC. Combined therapy is a new direction for the treatment of HCC, please refer to more literature.  (Please refer to: doi:10.4103/jcrt.JCRT_101_20; doi:10.4103/jcrt.JCRT_1848_20; doi:10.3389/fonc.2022.1029951).

Comments on the Quality of English Language

The English language is acceptable, and it is recommended to consider using technical terms and more concise expressions.

Author Response

Response to Editors and Reviewers Comments:

We would like to thank the editors and reviewers for their constructive assessment of our work and helpful recommendations. Please find our responses to your comments below. All comments are addressed in an individual manner.

Reviewer 2:

We have carefully completed the preview of this manuscript. This is a relatively complete review on the targeted therapy of HCC. Here are a few recommendations for this manuscript. At the part of Overview of Current Liver Directed Therapies for HCC.

  1. Firstly, because of the significant achievements in HAIC therapy in intermediate-stage HCC in recent years, I would suggest to add relevant content. (Please refer to: doi:10.1200/JCO.21.01963).

We thank the reviewers for their comments and agree.  We have therefore, added a new section discussing HAIC therapy and have included the reference.           

“Hepatic artery infusion chemotherapy (HAIC) and stereotactic body radiotherapy (SBRT) are locoregional techniques that have also been described in unresectable HCC (40). HAIC has been historically used in patients with unresectable HCC who have progressed after TACE or patients with portal vein thrombosis or large infiltrative tumors (40). Historically, the combination of 5-FU and cisplatin are drugs utilized (40). In a recent FOHAIC-1 trial, the combination of oxaliplatin and 5-FU was compared to the systemic agent sorafenib (41). Patients who received the combination therapy demonstrated significantly longer survival and were more likely to have their tumors downgraded. Hence, there is growing interest in using HAIC as an alternative for patients with advanced stage HCC who are not candidates for other locoregional therapies or who have not responded to systemic therapy (40).”

  1. Secondly, please consider to discuss the combination therapy of TACE and ablation.

We thank the reviewers for their comments and agree.  We have therefore, added the following:

“The combination of locoregional therapies is also an area of interest. In a meta-analysis, the combination of TACE and RFA compared to surgical resection showed no difference in overall survival but reduced complications in the combination therapy group (64).”  We have also added Figure 4 showing a case of combined cTACE and cryoablation.

  1. Thirdly, SBRT remains an important treatment option for HCC, please consider to add relevant content. At the part of Recent Advances and Innovations in Liver Directed Therapies for HCC.

We thank the reviewers for their comments and agree.  We have therefore, added a new section discussing SBRT.

“External beam radiotherapy originally had a limited role in the treatment of HCC due to the risk of radiation-induced liver injury (42). However, technological advancements have enabled higher doses to be delivered more precisely in SBRT. This form of radiation therapy has been shown to induce immunogenic cell death, which can have synergistic effects when combined with other therapies (42). It is generally used in early-stage HCC in patients who are not candidates for resection or ablation or as a palliative therapy in locally advanced or metastatic HCC (42). A study by Sapisochin et al. demonstrated similar survival and postoperative complication rates in patients with HCC who were bridged to transplant with either RFA, TACE, or SBRT (43). Further research is needed on the application of this therapy versus other locoregional therapies and its role in the HCC treatment paradigm.”

  1. Combined therapy is a new direction for the treatment of HCC, please refer to more literature.  (Please refer to: doi:10.4103/jcrt.JCRT_101_20; doi:10.4103/jcrt.JCRT_1848_20; doi:10.3389/fonc.2022.1029951).

Thank you for the thoughtful comments and suggestions for citations.  We added a table (table 1) that specifically addresses combination therapies. In addition, we have amended the text to include the following:

The first citation was added as a reference (See reference 62).

Though the second citation does not address combination therapy, we have updated the text to include this study.  We have added the following:
“Another study by Li et al. similarly showed a decrease in complications and prolonged interval between treatments with DEB-TACE (28).”

The third citation above has been added (See reference 61). 

Reviewer 3 Report

Comments and Suggestions for Authors

A well-organized review of the past, present, and future treatment of HCC. However, I believe that the review article would be enriched by a few additions.

First, it would be helpful to present a figure of each procedure in the percutaneous ablation part, such as figure 1,2.

Secondly, 3.2 combination therapies part is the most groundbreaking field at present, and it is meaningful to briefly table the characteristics and mechanisms of advanced stage drugs, RCTs, etc. In addition, the recent study (Camrelizumab plus rivoceranib versus sorafenib as first-line therapy for unresectable hepatocellular carcinoma (CARES-310): a randomized, open-label, international phase 3 study)) should be added.

Thank you.

Comments on the Quality of English Language

A well-organized review of the past, present, and future treatment of HCC. However, I believe that the review article would be enriched by a few additions.

First, it would be helpful to present a figure of each procedure in the percutaneous ablation part, such as figure 1,2.

Secondly, 3.2 combination therapies part is the most groundbreaking field at present, and it is meaningful to briefly table the characteristics and mechanisms of advanced stage drugs, RCTs, etc. In addition, the recent study (Camrelizumab plus rivoceranib versus sorafenib as first-line therapy for unresectable hepatocellular carcinoma (CARES-310): a randomized, open-label, international phase 3 study)) should be added.

Thank you.

Author Response

Response to Editors and Reviewers Comments:

We would like to thank the editors and reviewers for their constructive assessment of our work and helpful recommendations. Please find our responses to your comments below. All comments are addressed in an individual manner.

Reviewer 3:

A well-organized review of the past, present, and future treatment of HCC. However, I believe that the review article would be enriched by a few additions. 

  1. First, it would be helpful to present a figure of each procedure in the percutaneous ablation part, such as figure 1,2. 

We agree with the reviewers and have added Figure 1 (example of MWA) and Figure 4 (combined cryoablation and TACE).

  1. Secondly, 3.2 combination therapies part is the most groundbreaking field at present, and it is meaningful to briefly table the characteristics and mechanisms of advanced stage drugs, RCTs, etc. In addition, the recent study (Camrelizumab plus rivoceranib versus sorafenib as first-line therapy for unresectable hepatocellular carcinoma (CARES-310): a randomized, open-label, international phase 3 study)) should be added.

We agree with the reviewers and have updated the text to include this study.  The text now reads:
“Similarly, the combination of the immune checkpoint inhibitor Camrelizumab with the TKI Rivoceranib in the CARES-310 phase III trial showed significant increase in PFS and OS when compared to Sorafenib (53).”

Thank you.

Reviewer 4 Report

Comments and Suggestions for Authors

 Title: New Insights on Liver Directed Therapies in Hepatocellular Carcinoma

An overview of the most effective liver-directed treatments in HCC is given in this article.In addition to knowledge of more current developments in specific therapy and advanced techniques. Consider should be given to a few remarks.

1-    In the text, mention figures 1 and 2.

2-    List the chemotherapy regimens utilized, their combinations, and the outcomes for treating HCC in Table.

3-    Additional details regarding the effects of the chemotherapeutic treatments coated with nanoemulsion, including in vitro and animal experiments as well as human trials.

Author Response

Response to Editors and Reviewers Comments:

We would like to thank the editors and reviewers for their constructive assessment of our work and helpful recommendations. Please find our responses to your comments below. All comments are addressed in an individual manner.

Reviewer 4:

An overview of the most effective liver-directed treatments in HCC is given in this article. In addition to knowledge of more current developments in specific therapy and advanced techniques. Consider should be given to a few remarks. 

  1. In the text, mention figures 1 and 2. 

Thank you for pointing this out to us.  We have added references to these figures, which are now figures 2 and 3 due to the addition of figures 1 and 4 as discussed above. These references were added within the text to their corresponding sections.

  1. List the chemotherapy regimens utilized, their combinations, and the outcomes for treating HCC in Table.

We thank the reviewers for this recommendation, and have added Table 1 as an overview of the trials and retrospective studies focused on the combination of TACE and systemic therapies. Additional references were added to the table. We chose to focus on these combination therapies from the interventional radiology perspective as that is our area of expertise.

  1. Additional details regarding the effects of the chemotherapeutic treatments coated with nanoemulsion, including in vitroand animal experiments as well as human trials.

Thank you for this comment. There is a discussion regarding targeted drug delivery with nanoparticles in Section 3.3. The authors believe that the in vitro and animal experiments are outside the scope of this article. We have chosen to focus on the application of this technology in human trials. More information regarding the preclinical status of the nanoemulsion of chemotherapy was added to Section 3.3. The ThermoDox study was already included describes the most notable application of this technology thus far in human trials.  The amended text in section 3.3 now reads:

“Multiple preclinical trials in animal and cell models are focused on encapsulating chemotherapeutic agents into nanoparticles, with promising results in inhibiting tumor growth (69).”

Round 2

Reviewer 3 Report

Comments and Suggestions for Authors

I think the authors submitted meticulous responses and corrections.
Thank you for getting it resolved my requirement.  I don't have any additional comments or requests.

Thank you for your meticulous responses and corrections.